# Impact of Maize–Mushroom Intercropping on the Soil Bacterial Community Composition in Northeast China

**Xiaoqin Yang** [1], **Yang Wang** [1,*], **Luying Sun** [1], **Xiaoning Qi** [1], **Fengbin Song** [1] and **Xiancan Zhu** [1,2,*] 

1   Northeast Institute of Geography and Agroecology, Chinese Academy of Sciences, Changchun 130102, China; yangxiaoqin@iga.ac.cn (X.Y.); sunluying@iga.ac.cn (L.S.); qxn@iga.ac.cn (X.Q.); songfb@iga.ac.cn (F.S.)
2   College of Life Sciences, Anhui Normal University, Wuhu 241000, China
*   Correspondence: wangyang@iga.ac.cn (Y.W.); zhuxiancan@ahnu.edu.cn (X.Z.)

**Abstract:** Conservative agricultural practices have been adopted to improve soil quality and maintain crop productivity. An efficient intercropping of maize with mushroom has been developed in Northeast China. The objective of this study was to evaluate and compare the effects of planting patterns on the diversity and structure of the soil bacterial communities at a 0–20 cm depth in the black soil zone of Northeast China. The experiment consisted of monoculture of maize and mushroom, and intercropping in a split-plot arrangement. The characteristics of soil microbial communities were performed by 16S rRNA gene amplicom sequencing. The results showed that intercropping increased soil bacterial richness and diversity compared with maize monoculture. The relative abundances of Acidobacteria, Chloroflexi, Saccharibacteria and Planctomycetes were significantly higher, whereas Proteobacteria and Firmicutes were lower in intercropping than maize monoculture. Redundancy analysis suggested that pH, $NO_3^-$-N and $NH_4^+$-N contents had a notable effect on the structure of the bacterial communities. Moreover, intercropping significantly increased the relative abundance of carbohydrate metabolism pathway functional groups. Overall, these findings demonstrated that intercropping of maize with mushroom strongly impacts the physical and chemical properties of soil as well as the diversity and structure of the soil bacterial communities, suggesting this is a sustainable agricultural management practice in Northeast China.

**Keywords:** 16S rRNA; planting pattern; soil chemical properties; soil microbial community

## 1. Introduction

Maize (*Zea mays* L.) is not only one of the three major food crops in the world, but also a high-quality animal feed and an important industrial raw material. Maize production in China accounts for 21.9% of total maize output in the world. The northeast area is the largest maize producer in China, particularly in the Jilin province, which accounts for 1/8th of maize production in China [1]. However, the maize planting pattern in Northeast China was mainly based upon conventional agricultural practices (e.g., monoculture cropping, tillage and removal of crop residues) that have caused soil erosion, fertility decline, and loss of soil biodiversity [2]. Therefore, conservation and sustainable agricultural practices, such as intercropping, no tillage and mulching, have been adopted to improve soil quality and increase crop productivity.

Intercropping, a method of simultaneously planting two or more crops on the same field, has been practiced in many countries for several decades [3]. It increases the productivity per unit of land through better utilization of resources, control of disease and reducing soil erosion [4]. Soil microbial

activity, nutrient cycling, decomposition of organic matter, and physical and chemical properties can be changed in intercropping systems [5,6]. Soil organic matter, reflecting the physical, chemical and biological properties of soil, is an effective indicator of soil quality [2,4], which is defined as the capacity of a soil to function within ecosystem and land-use boundaries to sustain biological productivity, maintain environmental quality, and promote plant and animal health [7]. Mushroom compost is used as a soil conditioner for the growth of soybean, and the substrate pH for growing soybean is increased from acidity to neutrality [8]. The residual fungi and a large number of mycelium are incorporated into the soil, making the soil loose, which improves the physical and chemical properties of the soil, and increases the content of various nutrients in the soil in the maize–mushroom intercropping system [9]. Soil microorganisms play crucial roles in soil ecosystem functioning, as they are involved in soil nutrient cycling and energy flows, micro-ecology regulation and soil sustainable productivity [4,10]. In the soil ecosystem, soil bacteria are crucial decomposers that metabolize organic matter by secreting specific extracellular enzymes to break down large organic molecules into monomers, which are then available for plant uptake [11,12]. The cultivation of continuous monoculture often results in the accumulation of soil-borne pathogens, the growth inhibition of beneficial bacteria [13], and a decrease of bacterial diversity, ultimately [14]. Pear/mushroom intercropping significantly increased the number and biomass of cultivable microorganisms in the 0–40-cm soil layer and had a marked effect on the soil fertility of the pear garden and the quality of the pear fruit [15].

Oyster mushroom (class Basidiomycetes) is a fungus that is an excellent source of human nutrients (e.g., vitamins, minerals and micro nutrients) and has medicinal values [16]. It can be used as an intercropping crop to promote the efficient uptake of nutrients and resources [8]. For example, the mushroom (*Plearotus* spp.) intercropped with field-grown faba bean (*Vicia faba* L.) increased the faba bean dry seed yield and mushroom basidiocarp yield compared to sole cultivation [16]. Maize–mushroom intercropping systems have been previously studied and were proved to improve maize yield and land equivalent ratio [9,17]. However, few studies have attempted to investigate the effects of maize–mushroom intercropping on soil physical and chemical properties and the diversity and composition of soil microbial community. Therefore, the objective of this study was to explore the effect of maize–mushroom intercropping on the soil bacterial community in Northeast China. Recently, we developed an integrated agricultural management practice involving wide–narrow-row spacing alternation (160 + 40 cm), adoption of no tillage, and mulching by crop residues in maize cropping system in the Northeast China plain, resulting in an improvement of soil quality, reduced soil erosion, and a more sustainable use of cultivated land [18]. Furthermore, in order to use land resources with high efficiency, control disease, and increase the income of farmers, maize–mushroom intercropping was developed based on the integrated agricultural practice. We hypothesized that the intercropping could lead to (i) a marked effect on the structure and diversity of the soil bacterial community and (ii) an improvement of soil quality compared with monocultures.

## 2. Materials and Methods

### 2.1. Site Description and Experimental Design

The integrated agricultural management practice was initiated in spring 2012 and the maize–mushroom intercropping under integrated agricultural practice experiment was initiated in spring 2014 at Changchun Agricultural Experimental Field of the Northeast Institute of Geography and Agroecology, Jilin Province, China (43°59′54″ N, 125°23′57″ E). The climate is a north temperate continental monsoon. The mean annual air temperature is 7.2 °C. The mean annual precipitation is 530.5 mm. The frost-free period is 138 days. Snow cover usually occurs from November to April. The soil is a typical silty clay loam (classified as Mollisols), which is highly fertile, inherently productive and suitable soil for cultivation in the world [19], with organic matter 3.1%, total nitrogen 1.49 g kg$^{-1}$, total phosphorus 0.59 g kg$^{-1}$, total potassium 21.53 g kg$^{-1}$, available nitrogen 204.54 mg kg$^{-1}$, available

phosphorus 9.43 mg kg$^{-1}$, effective potassium 125.84 mg kg$^{-1}$, soil bulk density 1.19 g cm$^{-3}$, cation exchange capacity 23.3 cmol kg$^{-1}$, and pH 6.71.

The experiment was conducted in a plot comparison experiment with three treatments including (1) maize–mushroom intercropping, (2) maize monocultures, and (3) mushroom monocultures. Each plot was 20 m long and 8 m wide with four plot replicates for each treatment. The cultivars were maize Liangyu 99 and oyster mushroom (provided by Liaoning Sanyou Agricultural Biotechnology Company). The maize seeds were sown in late April and harvested in late September, and mushroom sticks were planted in early July and harvested in late September every year. In the maize monoculture treatment, the wide–narrow row spacing was 160 and 40 cm, respectively; no tillage was implemented; the maize stalks were cut at 35 cm above the ground, and then maize stalks were left in the field. Maize was planted on a narrow line of 40-cm spacing, with a plant spacing of 15 cm and a planting density of $6.5 \times 10^4$ plants ha$^{-1}$. The intercropping of maize with mushroom was based on the integrated agricultural management practice. In the intercropping and maize monoculture, maize seeds were sown in 40-cm, narrow rows using a small jukebox in late April. During the maize spinning period, a 10-cm-deep and 40-cm-wide sulcus was dug in advance to place mushroom sticks in the intercropping. Then, mushroom sticks were evenly placed in the sulcus, covered in 2–3 cm soil (Figure 1). The distance between mushroom and maize, mushroom and high stubble both were 27 cm. The mushroom row spacing of mushroom monoculture treatment was 160 cm. A based controlled release compound fertilizer (resin coating) of 600 kg ha$^{-1}$ (total nutrient ≥ 53%, Zn ≥ 2%) was applied to all treatments (as the base of the fertilizer) before rotating the soil (using the small rotating machine). The type of controlled release nutrient was nitrogen, and the amount of controlled release nutrient was greater or equal to 8%. In mid-May, the compound fertilizer was added once. Throughout the growth period of mushroom, intermittent spray irrigation was used to maintain the optimum moisture for mushroom production.

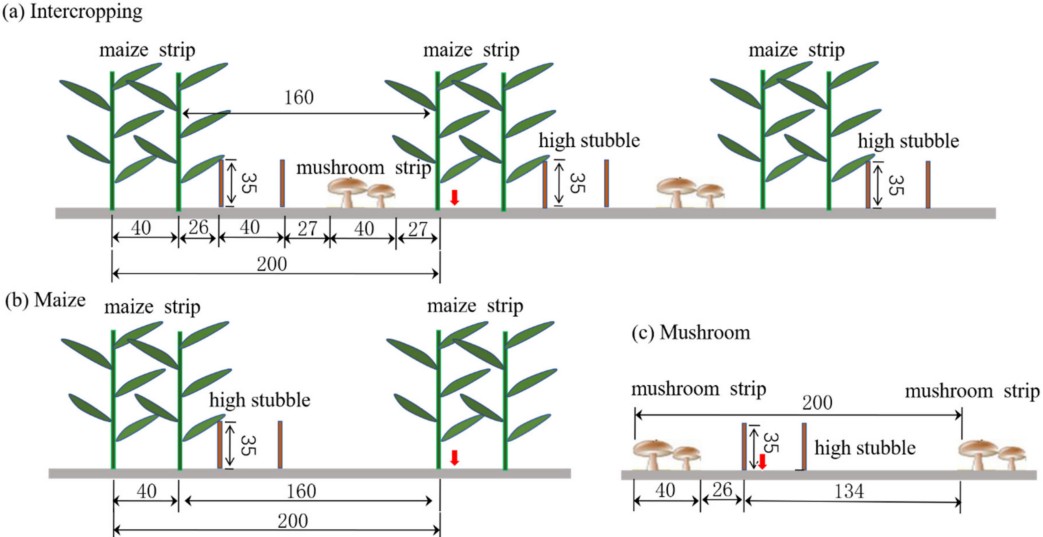

**Figure 1.** Schematic illustration of row placement of (**a**) maize–mushroom intercropping, (**b**) maize monoculture and (**c**) mushroom monoculture. Red arrows indicate the sampling point. The unit of distances is the centimeter (cm).

## 2.2. Soil Sampling

On September 20 2017 (After the mushrooms had been planted for 70 days), soil core samples (5.5 cm in diameter, 20 cm in depth, repeated four times) from each treatment were taken randomly from the maize rows (10 cm from the maize rows) of maize monoculture and intercropping treatment, and high stubble rows (10 cm from high stubble rows) of mushroom treatment. In total, there were 12

soil samples used for the analysis. All samples were sieved by a 2-mm sieve to remove rocks and were then separated into two parts: one part air dried to determine soil properties, and another stored at $-80\ °C$ for DNA extraction.

### 2.3. Soil Chemical Assays

Total nitrogen (TN), nitrate nitrogen ($NO_3^--N$), ammonium nitrogen ($NH_4^+-N$), were determined by a continuous flow analyzer (San++, Skalar, Breda, Holland). Soil organic matter (SOM) was measured by the method of potassium dichromate heating oxidation-volumetric based on the standard LY/T1237-1999 [20]. Briefly, 0.2 g of air-dried soil was added into 5 mL $K_2Cr_2O_7$ (0.8 M) and 5 mL concentrated $H_2SO_4$ solution and heated on a hot plate (300 °C) for 5 min. Three to four drops of phenanthroline indicator were then added, and titrated with $FeSO_4$ (0.4 M).

TN, available nitrogen (AN), $NH_4^+-N$, and $NO_3^--N$ were measured by standard methods based on LY/T1228-2015. For TN, 1 g soil was added to 1.8 g catalyst (selenium powder: copper sulfate: potassium sulfate = 1:10:100), mixed with 4 mL concentrated sulfuric acid, and removed on the electric heating plate until the soil became grayish white. The resulting solution was green and was transferred to a 100-mL volumetric flask after cooling. The solution was then diluted to volume with distilled water and shaken for testing. For AN, 2 g soil was mixed with 3 mL of 2% $H_3BO_3$, 10 mL of 1 M NaOH solution, and then incubated at 40 °C for 24 h. After cooling, the sample was titrated with HCl (0.012 M) until the color changed from blue-green to purple. For $NH_4^+-N$ and $NO_3^--N$, 5 g soil was mixed with 25 mL of KCl (2 M), shaken for 30 min, and filtered for testing.

To measure soil pH, 10 g soil was placed in a 100-mL beaker. Then, 50 mL of distilled water (water:soil = 5:1) was added, shaken for 30 min, and measured with a calibrated pH meter (Mettler-Toledo FE 20, Zurich, Switzerland).

### 2.4. DNA Extraction and MiSeq Sequencing

DNA was extracted from 0.25 g fresh soil using E.Z.N.A Mag-Bind Soil DNA Kits (OMEGA, Irving, TX, USA) according to the manufacturer's instructions. The DNA extraction quality was measured by 0.8% agarose gel electrophoresis, and the DNA was quantified by an ultraviolet spectrophotometer. The PCR amplification used Q5 high-fidelity DNA polymerase (NEB, Ipswich, MA, USA), and the V3-V4 region of the 16S rDNA genes were amplified using the primers 338F (ACTCCTACGGGAGGCAGCA) and 806R (GGACTACHVGGGTWTCTAAT) [21]. Referring to the preliminary quantitative results of gel electrophoresis, the PCR amplified product was subjected to fluorescence quantification, the fluorescent reagent was Quant-iTPicoGreen dsDNA Assay Kit, and the quantitative instrument was Microplate reader (BioTek, FLx800). A total of 20 pM DNA for each sample were pooled and sequenced in an Illumina MiSeq platform (Illumina, SanDiego, CA, USA) with a 600-cycle kit (2 × 300 bp paired ends). The Miseq sequencing raw data were deposited in the NCBI Sequence Read Archive database, and the project ID is PRJNA432129 and BioSample is SAMN10362720.

### 2.5. Data Analysis

The original paired-end sequencing data were exported in a FASTQ format. Sequences were removed if the read length was <150 bp, with a mean quality score <20 [22]. Sequence analysis was performed using QIIME software (Version 1.8.0) [23]. Sequences with ≥97% similarity were assigned to the same operational taxonomic units (OTUs) using UCLUST [24]. In detail, the sequences were merged according to ≥97% similarity into OTUs, and the most abundant sequence in each OTU was chose as the representative OTU sequence. Then, according to the number of sequences of each OTU in each sample, a matrix file of OTU abundance in each sample was constructed (i.e., OTU table). The relative abundance of OTUs with <0.001% of the total reads of all samples were removed [25]. The Greengenes database (Release 13.8) was used to annotate taxonomic information [26].

Data were compared using analysis of variance (ANOVA) in IBM SPSS 23.0 software (SPSS Inc., USA). Alpha diversity (Chao1 and Shannon index) was calculated with QIIME (Version 1.8.0).

Pearson's correlation coefficients between soil properties, OTU richness (Chao1 index) and diversity (Shannon index), and bacterial phyla were computed.

Differences of soil bacterial communities based on OTUs between treatments were analyzed using LEfSe [27]. Cluster analysis of soil bacterial communities based on the non-metric multidimensional scaling (NMDS) dissimilarity matrix was performed using QIIME. Permutational multivariate analysis of variance (PERMANOVA; function 'adonis') was adopted to compare community composition on three treatments with QIIME. Redundancy analysis (RDA) was carried out to explore the relationship between soil properties and microbial community composition using the 'vegan' package in the R program (Version 3.3.1). Function predictions were classified into KEGG pathways using PICRUSt (Version 1.1.4) method [28]. In detail, firstly, OTU table was standardized by copy number; the full-length 16S rRNA gene sequence of the tested microbial genome was used to infer the gene function spectrum of their common ancestor; then, the gene function profiles of other untested species in the Greengenes 16S rRNA gene full-length sequence database was inferred and constructed the gene function prediction profiles of the entire lineage of archaea and bacteria; thirdly, the 16S rRNA gene sequence data obtained by sequencing with the Greengenes database was compared to find the "nearest neighbor of the reference sequence" of each sequenced sequence and classified it as a reference OTU; the obtained OTU abundance matrix according to the copy number of the rRNA gene of the nearest neighbor of the reference sequence was corrected; finally, the microbial composition data to the known gene function profile database was "mapped" to realize the prediction of the metabolic function of the microbial communities. The KEGG pathways statistical analysis was implemented using SPSS.

## 3. Results

### 3.1. Soil Chemical Properties

The planting pattern was found to significantly affect soil pH and the contents of AN, $NO_3^-$-N, $NH_4^+$-N, and SOM (Table 1). The soil pH and SOM were markedly increased in intercropping, whereas the contents of AN, $NO_3^-$-N and $NH_4^+$-N were decreased, compared with maize monoculture. Soil pH was higher, whereas AN, $NO_3^-$-N and $NH_4^+$-N contents were lower in mushroom monoculture than maize monoculture.

**Table 1.** Soil chemical properties under maize monoculture, mushroom monoculture and maize–mushroom intercropping.

| Planting Pattern | pH | TN (g kg$^{-1}$) | AN (mg kg$^{-1}$) | $NO_3^-$-N (mg kg$^{-1}$) | $NH_4^+$-N (mg kg$^{-1}$) | SOM% |
|---|---|---|---|---|---|---|
| Maize | 6.37 ± 0.10 b | 1.28 ± 1.49 a | 257 ± 33.6 a | 191 ± 58 a | 51.3 ± 15.3 a | 3.12 ± 0.15 b |
| Mushroom | 7.49 ± 0.05 a | 1.39 ± 0.60 a | 169 ± 10.4 b | 2.19 ± 0.27 b | 4.15 ± 0.46 b | 3.23 ± 0.05 ab |
| Intercropping | 7.40 ± 0.04 a | 1.51 ± 1.05 a | 170 ± 8.65 b | 19.8 ± 3.39 b | 8.41 ± 0.69 b | 3.59 ± 0.10 a |

Values are means ± standard errors. Mean values in each column followed by the different letters are significantly different ($p < 0.05$) according to Tukey's test. The number of replicates per treatment is 4 ($n = 4$). TN, AN, $NO_3^-$-N, $NH_4^+$-N and SOM represent total nitrogen, available nitrogen, nitrate nitrogen, ammonium nitrogen and soil organic matter, respectively.

### 3.2. Soil Bacterial Community Diversity

Sequencing of the 16S rRNA gene fragment from 12 soil samples produced a total of 462,813 sequences (Table S1). After filtration, alignment, pre-clustering and removal of chimeric sequences and singletons, 359,341 sequences were obtained. They were assigned into 18,519 OTUs.

Intercropping and mushroom monoculture had higher bacterial OTU richness and Shannon index compared with maize monoculture (Figure 2). There was a striking positive relationship between Shannon index and pH and SOM, whereas there was a negative association between Shannon index and the contents of AN, $NO_3^-$-N, $NH_4^+$-N (Table S2).

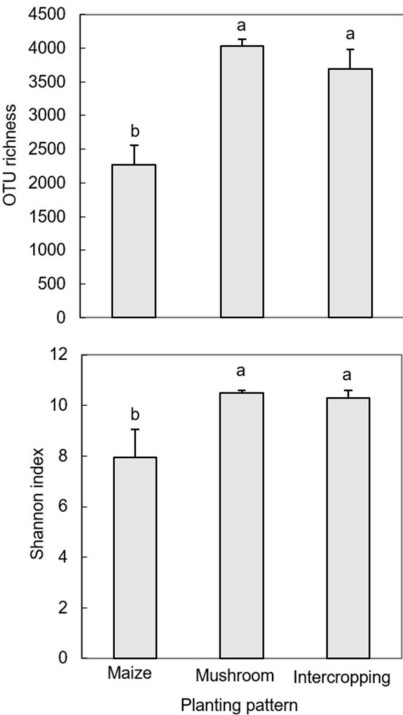

**Figure 2.** Operational taxonomic units (OTUs) richness and Shannon index of soil bacterial community under maize monoculture, mushroom monoculture and maize–mushroom intercropping. The error bars represent standard errors (SE). Different letters indicate significant different ($p < 0.05$) according to Tukey's test. The number of replicates per treatment is 4 ($n = 4$).

### 3.3. Soil Bacterial Community Structure

In order to characterize the effect of planting pattern on soil bacterial communities, the relative abundances at phylum, class, order, family and genus levels were analyzed. Then, 16S rRNA gene amplicon sequencing showed that the dominant bacterial phyla were Proteobacteria, Chloroflexi, Acidobacteria, Actinobacteria, Firmicutes, Gemmatimonadetes, Cyanobacteria, Saccharibacteria, Bacteroidetes, Nitrospirae, Planctomycetes, Verrucomicrobia and Parcubacteria (Figure 3), and these groups accounted for over 88–95% of the sequences. Moreover, soil chemical properties were closely correlated with the relative abundance of some dominant microbial phyla groups (Table S2). There was a marked positive association between soil pH and Chloroflexi, Acidobacteria, Planctomycetes, Verrucomicrobia, and Parcubacteria, whereas the relationships were negative between the contents of AN, $NO_3^--N$, $NH_4^+-N$ and those dominant microbial phyla.

In total, 13 phyla, 31 classes, 64 orders, and 84 families in the bacterial community were significantly affected by planting pattern. At the phylum level, planting pattern significantly altered the relative abundance of Chloroflexi, Acidobacteria, Saccharibacteria, Planctomycetes, Armatimonadetes, Fusobacteria, Euryarchaeota, WS6, Elusimicrobia, Peregrinibacteria, WWE3 and Gracilibacteria. Intercropping had higher relative abundance of Acidobacteria, Chloroflexi, and Planctomycetes, and lower relative abundance of Proteobacteria and Firmicutes compared with maize monoculture treatment (Figure 3). Intercropping had higher relative abundance of Saccharibacteria, and lower Cyanobacteria than mushroom monoculture. The distribution of related genera varied between maize monoculture, mushroom monoculture and intercropping soils. *Pseudomonas*, *Sphingomonas*, *Lactobacillus* and *Rhodanobacter* were the most abundant genera across all soil samples, representing 5.57%, 2.83%, 0.19%, and 3.74% of all classified sequences in intercropping, 17.24%, 9.28%, 10.79%, and 2.05% in maize monoculture and 6.64%, 1.59%, 0.14%, and 1.05% in mushroom monoculture, respectively. *Gemmatimonas*, *Cystobacteraceae*, *Marmoricola*, *Bradyrhizobium*, *Streptomyces*, *Rhodoglobus*, *Nakamurella*, *Frateuria*, *Sphingobium*, *Woodsholea*, and *Oribacterium* showed an increased relative abundance in

intercropping, while the relative abundance of *Streptomycetaceae* decreased in intercropping compared with maize monoculture (Figure 4).

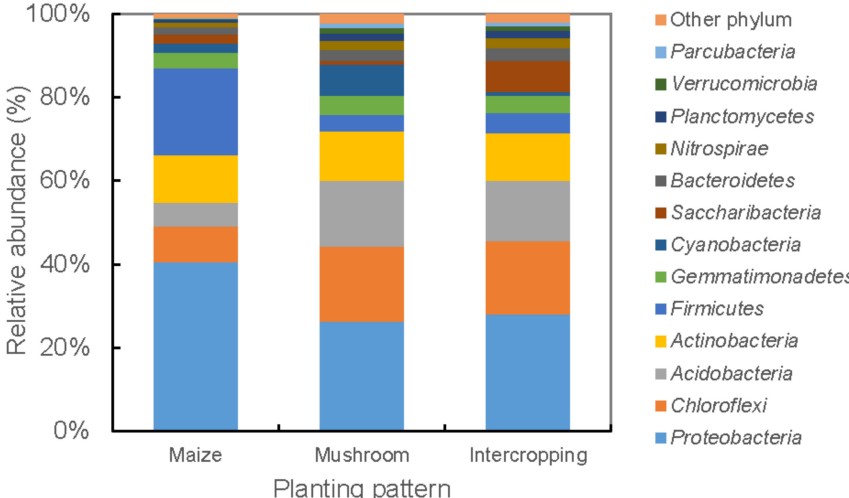

**Figure 3.** Relative abundances of the dominant phyla of bacteria at 0-20-cm soil depths under maize monoculture, mushroom monoculture and maize–mushroom intercropping. The number of replicates per treatment is 4 (*n* = 4).

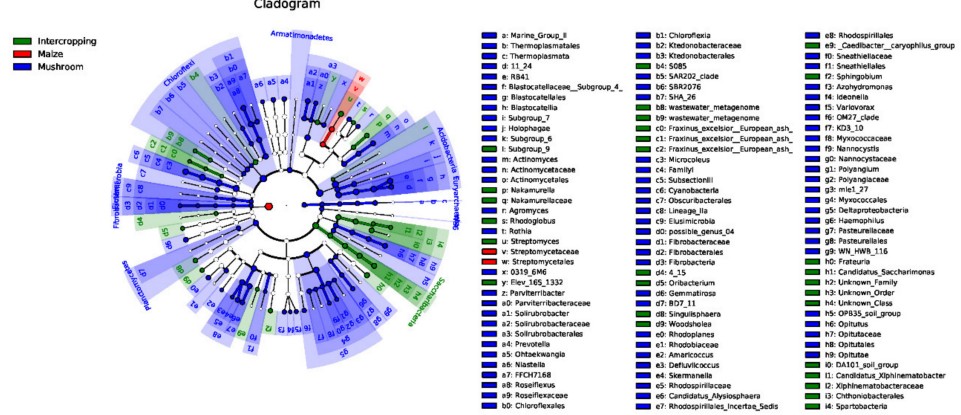

**Figure 4.** Cladogram of bacteria at 0–20-cm soil depths under maize monoculture, mushroom monoculture and maize–mushroom intercropping. The Cladogram shows the hierarchical relationship of all classification units from the phylum to the genus (from the inner circle to the outer circle). The node size corresponds to the average relative abundance of the classification units. The blue, green and red node represents that the difference of bacterial relative abundance between groups is significant. The letters identify the taxon name that has a significant difference between groups. The number of replicates per treatment is 4 (*n* = 4).

*3.4. Comparative Analysis of Soil Bacterial Community*

Based on the NMDS dissimilarity matrix, hierarchical cluster analysis for investigated the beta diversity of bacterial communities showed that soil bacterial communities were affected by planting patterns (Figure 5). PERMANOVA analysis indicated that soil bacterial community composition was significantly affected by maize monoculture, mushroom monoculture and intercropping treatments. The RDA was performed to determine the strength of the association between the soil bacterial community and soil physical, chemical properties. RDA revealed a strong difference between maize monoculture, mushroom monoculture and intercropping soils (Figure 6). The first two canonical axes

are responsible for 45.73% of variance (25.42% by RDA1 axis and 20.31% by RDA2 axis). The RDA indicated that pH, $NO_3^--N$ and $NH_4^+-N$ contents had an extremely significant influence on the structure of the bacterial community ($p < 0.05$) (Table S3).

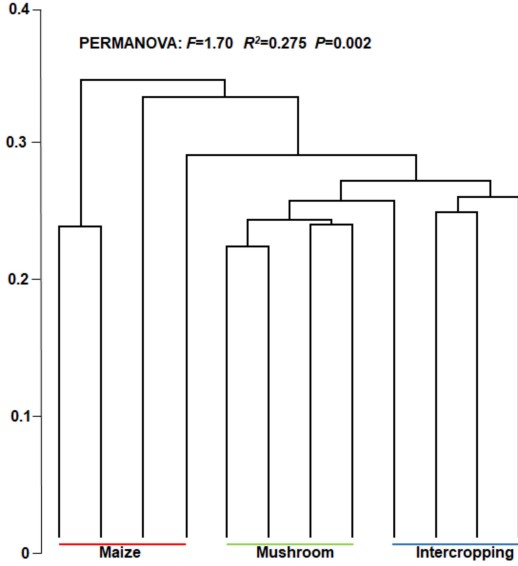

**Figure 5.** Hierarchical cluster analysis of soil bacterial communities based on the NMDS dissimilarity matrix among maize monoculture, mushroom monoculture and intercropping. Permutational multivariate analysis of variance (PERMANOVA) was adopted to compare community composition on three treatments. The number of replicates per treatment is 4 ($n = 4$).

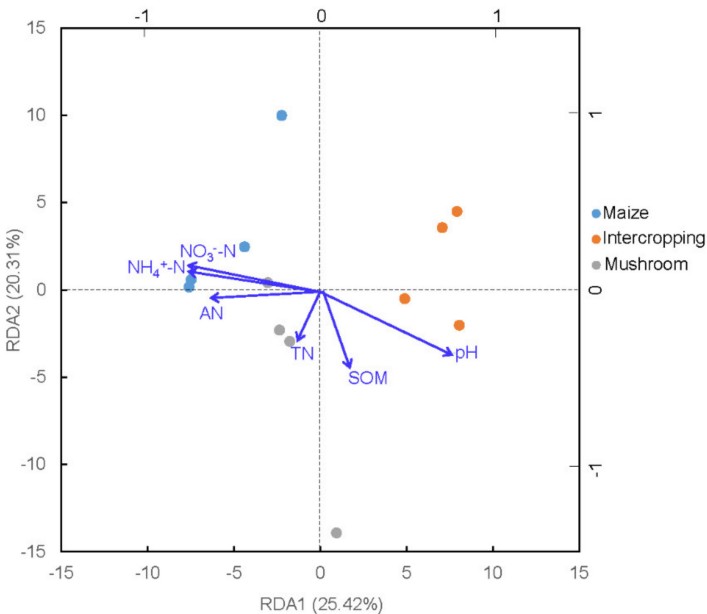

**Figure 6.** Redundancy analysis (RDA) of soil bacterial community structure and soil properties under maize monoculture, mushroom monoculture and intercropping. Soil factors indicated in blue text include pH, contents of organic matter (SOM), total nitrogen (TN), available nitrogen (AN), nitrate nitrogen ($NO_3^--N$) and ammonium nitrogen ($NH_4^+-N$). The circles are the RDA scores of the samples and the arrows are the scores of the soil variables by RDA. The number of replicates per treatment is 4 ($n = 4$).

### 3.5. Metabolism of Soil Microbial Community

Potential metabolism were assigned to predicted functional annotation of protein sequences. The KEGG metabolic pathway difference analysis of soil bacteria revealed significant changes in 11 metabolic networks between the groups of planting patterns. The analysis showed that intercropping significantly increased the relative abundance of carbohydrate metabolism pathway functional groups compared with maize monoculture (Figure 7). In addition, the relative abundance of glycan biosynthesis and other secondary metabolites functional groups in intercropping and mushroom monoculture was higher compared with maize monoculture ($p < 0.05$) (Figure 7 and Table S4).

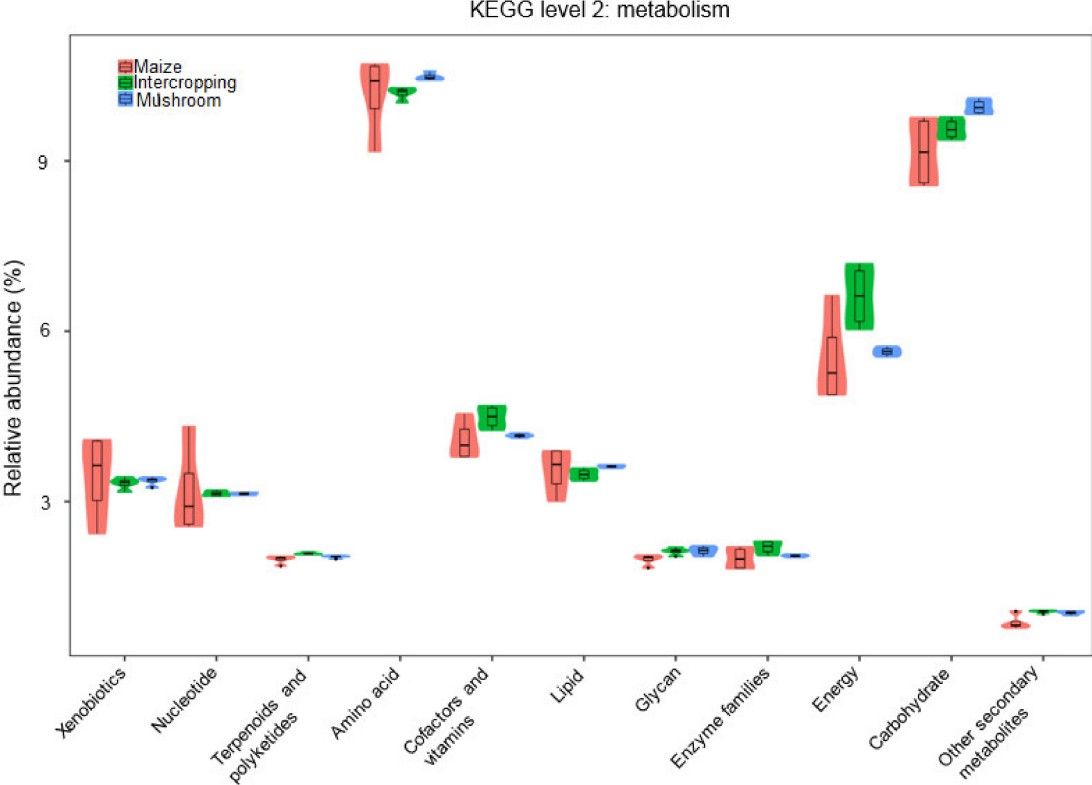

**Figure 7.** The KEGG metabolic pathway difference analysis of bacteria at 0–20-cm soil depths under maize monoculture, mushroom monoculture and intercropping. The number of replicates per treatment is 4 ($n = 4$).

## 4. Discussion

### 4.1. Soil Physicochemical Properties

Nutrients such as carbon, nitrogen, phosphorus and potassium, are essential for proper plant growth. Soil carbon and nitrogen contents are the most sensitive indicators of soil quality [29], and it has been suggested that below ground interspecific interactions improved soil nitrogen supply capacity [30], mobilization [31] and increased TN in intercropping [32]. The results of the current study have shown that contents of AN, $NO_3^-$-N, $NH_4^+$-N in intercropping treatment were no higher than maize monoculture treatment. Below ground interactions through intercropping could affect N-cycling [30,33]. Moreover, soil nitrate nitrogen is dynamic, and is influenced by soil particle distribution, soil depth, and precipitation, and varies during crop growth and development [34]. Soil nitrate would likely decline faster in intercropping and mushroom monoculture treatments in comparison with maize monoculture, mostly due to decomposition of mineral nutrients by saprotrophs and leaching losses [35], using intermittent spray irrigation to maintain the optimum moisture for

mushroom production. Vieira et al. [36] also showed that nitrogen losses after 10 and 15 days impacted by mushroom yield. In this study, intercropping treatment significantly increased the contents of SOM compared with maize monoculture. This is consistent with previous studies that demonstrated that soil organic carbon fraction, carbon pool management index and soil carbon sequestration were improved under intercropping [4,37,38].

### 4.2. Bacterial Community Diversity

The soil microbial community, a biomarker indicator of soil quality and ecosystem processes [39], is very sensitive to vegetation. On the other hand, it can also strongly affect plant growth and yield formation [40]. Our study revealed that intercropping significantly increased the OTU richness and diversity of the bacterial community compared to maize monoculture treatment. Consistent with our findings, Fu et al. [4] reported that maize–soybean intercropping had higher Shannon index compared with monocultures. Qin et al. [41] demonstrated that maize–potato intercropping increased the carbon source utilization rate and diversity of the microbial community. In addition, the bacterial Shannon index was correlated with contents of N, SOM and pH, suggesting that the improvements in C and N source utilization were beneficial to increased soil bacterial diversity, such as species richness [12]. Moreover, soil bacterial community could be changed by pH with a higher bacterial diversity in neutral soil than acidic soil, which a significant positive association between Shannon index and pH was found in this study.

### 4.3. Bacterial Community Structure

Microbial community composition has large effects on organic matter dynamics and nutrient cycling, and can influence soil function and ecosystem sustainability [42]. In total, 46 phyla, 168 classes, 356 orders, 605 families and 1207 genera of bacterial communities were obtained in our samples. Within the thirteen dominant bacterial phyla, Proteobacteria was the most abundant bacterial phylum, which was consistent with the results of mulberry and alfalfa intercropping system [43]. The major microbial phyla, such as Chloroflexi, Acidobacteria, Actinobacteria, identified in this study are often observed in other soils, though the relative abundance was different [44,45].

Previous studies have indicated that the dominant bacterial phyla could be changed by manipulating planting patterns and plant species [4,12,43]. In the present study, the bacterial community structures among intercropping and maize and mushroom monoculture treatments were significantly different. Planting patterns significantly affected the four dominant and eight other bacterial phyla. The relative abundance of Chloroflexi, Acidobacteria, Planctomycetes and Saccharibacteria were higher in intercropping than maize or mushroom monoculture treatments. Anaerolineae, a dominant class of Chloroflexi, has been thought to be ubiquitous and to play important roles in ecosystems [46]. The greater abundance of Chloroflexi in the intercropping than maize monoculture treatment likely indicated that intercropping would better coordinate soil ecosystems [47]. Acidobacteria is an acidophilic and oligotrophic chemoorganotrophic bacterium; Planctomycetes play possible role in the evolution of the methane cycle [48]. Saccharibacteria play a role in the degradation of various organic compounds as well as sugar compounds under aerobic, nitrate reducing, and anaerobic conditions [49]. Although the phylum Proteobacteria was not affected by planting pattern, Gammaproteobacteria, the most dominant class of Proteobacteria in our study, had a prominent higher abundance under intercropping than maize monoculture treatment, stimulated by higher SOM and lower nitrate nitrogen contents of intercropping treatment [50]. *Pseudomonas* is one of the widely distributed plant growth promoting rhizobacteria [51]. The result revealed that the soil *Pseudomonas* community was affected by planting pattern. Consistent with previous studies, the altered plant species affected the microbial community composition [52].

In this study, the relative abundance of Actinobacteria in intercropping and maize monoculture treatment were significant different at class level. At genus level, *Gemmatimonas*, *Streptomyces*, *Nakamurella* and *Frateuria* had greater abundance in intercropping soils. *Actinobacteria* have a critical

role in decomposition of soil organic materials, such as cellulose, chitin and polysaccharides [53]. Streptomyces could produce bioactive secondary metabolites which show antifungals and antivirals biological activities [54], and have more efficient secretion mechanisms which could promote protein solubilization [55]; *Nakamurella* is able to accumulate polysaccharides [56]; *Frateuria* is able to enhance potassium uptake efficiently in plants and has been found to increase biomass and nutrient content [57]. In addition, *Gemmatimonas*, the dominant genus of Gemmatimonadetes, is involved in modulating carbon and nitrogen intake, decomposing polyaromatic carbon and promoting plant development [58]. The greater abundance of *Actinobacteria*, *Streptomyces*, *Nakamurella*, *Frateuria* and *Gemmatimonas* likely be attributed to the effects of secondary metabolites such as carbohydrates, free amino acids, and nucleotides produced by mushroom during metabolism [59]. Our study showed that the relative abundance of glycan and secondary metabolites functional groups in intercropping and mushroom monoculture significantly increased compared with maize monoculture. This is consistent with the other research that intercropping lead to changes in plants accumulation of minerals and secondary metabolites [60].

To investigate the relationships between soil microbial community structure and measured soil variables in the maize monoculture, monoculture mushroom and intercropping systems, we analyzed the dominant bacterial phyla and OTUs data using Pearson's correlation and RDA. The soil variables have a substantial impact on the dominant bacterial phyla. In this study, SOM, TN, AN, $NO_3^-$-N, $NH_4^+$-N, and pH had positive/negative correlations with the dominant bacterial phyla. For example, the abundance of Chloroflexi and Planctomycetes were positively correlated with SOM content, which indicated that SOM correlated with the relative abundance of these bacteria [4]. SOM may play non-negligible roles in influencing on the soil microbial community structure by affecting the metabolism of soil microbes [61]. The output of RDA with soil variation in bacterial community indicated that pH plays critical roles in the structure of the bacterial community. This was in accordance with other studies that indicated that soil pH was a major factor in determining the structure of the soil bacterial community [12,61].

## 5. Conclusions

In total, 13 phyla, 31 classes, 64 orders, 84 families in the bacterial community were significantly affected by planting pattern. The results revealed that SOM, TN contents, Shannon index and the relative abundance of Chloroflexi, Acidobacteria, Saccharibacteria, Planctomycetes and carbohydrate metabolism pathway functional groups were significantly increased in intercropping compared with maize monoculture treatment. Moreover, soil chemical properties closely correlated with OTU richness, Shannon index and the relative abundance of Chloroflexi, Acidobacteria, Planctomycetes, Verrucomicrobia, Parcubacteria phyla groups. Our study demonstrates that intercropping of maize with mushroom affected the physical and chemical properties of the soil, and altered the structure and diversity of the soil microbial community. These results suggest that this crop production system could be a sustainable efficient agricultural management practice in Northeast China.

**Supplementary Materials:** The following are available online at http://www.mdpi.com/2073-4395/10/10/1526/s1, Table S1: Numbers of bacterial sequences and OTUs identified by 16S rRNA gene sequencing, Table S2: Pearson's correlation coefficients between relative abundances of dominant bacterial phyla and soil properties, Table S3: The environmental vectors onto two ordination of redundancy analysis, Table S4: The analysis of variance of the KEGG metabolic pathway.

**Author Contributions:** Conceptualization, F.S. and X.Z.; methodology, X.Y.; software, X.Y. and X.Z.; investigation, X.Y., L.S. and X.Q.; data curation, X.Y.; writing—original draft preparation, X.Y.; writing—review and editing, Y.W. and X.Z.; supervision, X.Q.; project administration, Y.W. and X.Z. All authors have read and agreed to the published version of the manuscript.

**Funding:** This research was funded by the "One-Three-Five" Strategic Planning Program of Chinese Academy of Sciences, grant number IGA-135-04.

**Conflicts of Interest:** The authors declare no conflict of interest.

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
