# Peer review of "Impact of Maize–Mushroom Intercropping on the Soil Bacterial Community Composition in Northeast China"

_agronomy, doi:10.3390/agronomy10101526_

Round 1
Reviewer 1 Report
Dear authors,
I have carefully read the manuscript. The topic is interesting and might have some application. The authors have done considerable work on data processing. However, the results and the recent literature are not sufficiently utilized to draw important conclusions. The discussion seems to be a review of the results, lacking of references, and scientific justifications. Below, I subjoin the specific comments to the manuscript.
Title
I suggest “Impact of maize-mushroom intercropping on the soil bacterial…” as a title for the article.
Abstract
Please, change the “monoculture maize/mushroom” to “maize/mushroom monoculture” or “monoculture of maize/mushroom” through all text
L18: Please, change “was” to “were”
L25: Please, change “physiochemical properties” to “physical and chemical properties”
Introduction
L39-45: I would prefer these sentences to be part of the last paragraph, with the description of the objective of the study. Please, rewrite the lines 39-42 and check for grammar “…resulting in improve..., reduce…, and achieving…”
L52-53: Please, rewrite the sentence. Use the present tense and check the expressive language.
L60-62: I consider that the words “may” and “could” are not appropriate to be used in the introduction. Please, check the reference, it does not seem to support the sentence.
L71: Please, rewrite. It is not clear that the literature, that concerns this issue, is restricted. Add references to support the sentence.
Materials and Methods
The description of the experimental design needs to be improved. The authors have to give more information, as detailed below. Also, the grammar has to be checked.
L84: Please, explain what is the ”black soil”. What are its characteristics? The soil texture is not mentioned.
L88: Please, add “a” before “plot comparison”.
L90: It is hard to understand what you mean. Do the replicates concern the treatments or the soil samples?
L91: Please, make clear what did you planted (seeds, seedlings, etc), i.e. “the maize seeds were…” and how.
L93: How long the field has been cultivated?
L97: The auxiliary verb is missing.
L97-100: It is not clear enough. Please, be more interpretative.
L102: It is necessary to provide more information about the application of fertilizer, as it can affect soil properties. When it was applied, it was applied to all treatments, what were its components, how and how often it was applied, etc.?
L108-111: Please, rewrite the sentence. The soil sampling in mushroom monoculture is not clear enough.
L112 and L128: I suppose it is not suitable to use air-dried soil samples for measuring NH4-N, NO3-N because these are changed during drying.
L115: Please, add “and” before “… ammonium nitrogen…” and change the auxiliary verb to “were”.
L152: Reference is needed here.
L154: Why did you choose to use a non-parametric test and not the ANOVA for the data analysis? Like most non-parametric tests, the Kruskal – Wallis test is not as powerful as the ANOVA.
L158: You have already mentioned the statistic software that you used, in L153.
L166: Please, replace were to was.
Results
1. The authors need to explain the reason that they used each analysis in a sentence (for example an RDA was performed to determine the strength of the association between the soil organic C and …). Also, authors have to be more detailed while they describe the results.
L168: Please, check the values of available nitrogen, nitrate-nitrogen, and ammonium-nitrogen (see also in Table 1).
L217: I did not find in figure 4 the Arthrobacter and Azospirillum for maize monoculture.
L218: What do you mean with “an opposite trend”?
L234-235: How did you come up with this result? Please, be more explicable/detailed.
L255: And what about the mushroom monoculture?
Discussion
1. In lines 261-273, you quoted the results of your study and previous studies, but there is no scientific interpretation of your results. For example, “The results of the current study have shown that contents of AN, NO3--N, NH4+-N in intercropping treatment were no higher than monoculture maize treatment.”, why do you think it happened? Also, there is no report for Total Nitrogen (TN), why it has not been affected by the treatments?
2. In lines 296-312, you refer to the roles of bacterial phyla and genus but there is no connection with the results and authors do not give explanations. For example, in line 323, you refer that some genus were abundant in the intercropping system, but you do not try to explain this result, why they were abundant, what could be a possible explanation on this, what is remarked in other similar studies?
3. Also, in the text, there are no explanations/possible scenarios on the no-differences (according to results) between the intercropping system and mushroom monoculture.
L269-270: The sentence is not clear enough. I do not understand it.
- “Soil nitrate would likely decline faster than in intercropping…” in comparison with what?
- “…mostly due to uptake by the maize crop and leaching losses” that occur in the intercropping system or in maize monoculture?
L282-285: Please, the lines are not understandable and need to be changed
L286: Which soils are considered acidic and neutral in this study?
L296: Please, use the plural form of “study”’.
L304: Is there any reference for this?
L310-312: It is hard to follow… Gammaproteobacteria, contributed to higher SOM and lower nitrate-nitrogen contents in the intercropping system that stimulated the activities of Gammaproteobacteria?
Conclusions
1. How did you conclude that the maize-mushroom intercropping system improved soil quality? The only parameters that seem to be increased in the intercropping system in comparison to maize monoculture, are pH and SOM. The conclusion is not well-supported.
L347: I believe that the word “strong” in the text, is exaggerated.
L348: Please, delete this phrase. It is repeated in the following sentence.
L349: Please, delete “also”.
Tables and Figures
1. Please, remove from all legends “at 0-20cm soil depths”, add the number of replicates per treatment in legends of tables and figures (n=?), and check for grammar.
Table 1: Check the values of available, nitrate, and ammonium nitrogen. They seem to be very different between treatments. Please, replace “Practices” with “Planting pattern”, as you use this in the text. Please, add “respectively” at the end of the footnote.
Figure 1: The schematic illustration is quite helpful. However, there are maize stalks (high stubble) on the image of mushroom monoculture; what for? Also, in the legend, change “The units is cm”, I suggest to be replaced by: “The unit of distances is the centimeter (cm)”
Figure 4: It has no meaning the phrase in the legend “…and higher relative abundance”.
Reviewer 2 Report
The manuscript titled “Impact of intercropping of maize with mushroom on the soil bacterial community composition in Northeast China” deals with the description of the bacterial soil community after maize-mushroom intercropping based on a 16S rRNA gene amplicon sequencing. Intercropping is a sustainable agricultural practice that might improve soil quality and field yields. Authors presents clearly an analysis of how this agricultural practice may benefit to bacterial soil richness and diversity.
The manuscript is very well organized. However, the metagenome analysis and the manuscript can be improved.
Specific comments:
- Lines 17-18: I suggest including the metagenome sequence method of this study: 16S rRNA gene amplicon sequencing
- Line 60: "uptake [11,12]"
- Lines 67-68: "efficient uptake of nutrients and resources [8]. For example, the mushroom (Plearotus)…."
- Lines 84-88: These are soil properties before the analysis? Did you measure them, or these data came from some previous analyses? Please detail it or reference it.
- Lines 101-102: Please, detail the composition or the base of the fertilizer, if possible.
- Line 111: Please, detail the total number of samples used for the analysis
- Line 121: and NO3- -N
- Line 145: Regarding SRA submission of sequences: please, find that the description of each sample (e.g. https://www.ncbi.nlm.nih.gov/sra/SRX4968709[accn]) have a different value for the 16S rRNA gene. The number "16" from 16S rRNA gene is changed in many of the samples. Please, correct this. In addition, it seems that you have at least another experiment under this bioproject number. Therefore, I would recommend detailing the origin of each sample (maize or mushroom and the plot) either in the manuscript or in the NCBI, including the sampleID.
- Line 149: add reference for QIIME (CAPORASO, J. Gregory, et al. QIIME allows analysis of high-throughput community sequencing data. Nature methods, 2010, vol. 7, no 5, p. 335-336).
- Line 150: please, detail if you employed an open-reference or a close-reference method for OTU picking.
- Line 151: please, provide the Greengenes database version.
- Line 160: add reference for LEfSE
- Line 165 and 166: add version and reference for PICRUST. In the case that you have version 1 installed instead of version 2: did you choose a closed reference method of OTU picking for functional predictions? In the case of you used an open-reference method for general taxonomic assignment and a closed-reference method for metagenome predictions, please, specify it. In the case of you employed an open-reference method before PICRUST analysis, please, repeat it with a closed-reference method. For more info, see the third and four paragraphs from: Douglas, G.M., Maffei, V.J., Zaneveld, J.R. et al. PICRUSt2 for prediction of metagenome functions. Nat Biotechnol 38, 685–688 (2020). https://doi.org/10.1038/s41587-020-0548-6. In the case you have version 2 installed, just specify the OTU picking method, but do not change anything.
- Line 166: Did you normalize the OTU table by copy number before performing metagenome predictions?
- Line 166: “using R”
- Line 175: Please, adjust the text content to the cell size, do not break words
- Line 184: Figure 2, OTU richness graphic: the “0” value is partially deleted, please correct it.
- Section 3.3.: It is possible that the soil had some vegetal content. Therefore, your total DNA might have some plant chloroplast DNA. 16S rRNA primers may have amplify that gene, and the Greengenes database may classify those sequences as “class Chloroplast”. In the Table S3 you will find that you have 0.16 – 1.18% of the sequences classify as chloroplast. This could obscure the analyses. Therefore, you should discard those sequences. You can do that through the following qiime script: py -i input_otu_table.biom -o otu_table_without_chloroplast.biom -n c__ Chloroplast
The same with the order “Fraxinus_excelsior_(European_ash)”, I would remove it.
In the case you change the OTU table, I guess you should repeat subsequent analyses with these new data.
- Line 194: Change Prcubacteria to Parcubacteria
- Lines 196-197: Table S1 instead of table S2?
- Line 200: Could you increase the quality of the figure 4 for the final version of the manuscript?
- Line 243: “20 cm”
- Line 312: I would add some specific comments regarding the role of Pseudomonas on soils or microbe-plant interactions. Since Pseudomonas is the predominant genus, it may be relevant.
- Line 313 and 315: Please, change Actinobacteria to Acidobacteria in the case that you want to write about this class. If the correct is Actinobacteria, please, change the reference 42, which is related to Acidobacteria.
- Line 317: “antifungal and antiviral biological activities”
- Lines 324-326: How can the effect of those molecules influence nutrient intake and stimulate carbon and nitrogen cycles? Do you mean that the availability of those nutrients will be used for microbes for their nutrition? Please, clarify the sentence
- Line 333-334: You did not prove if SOM has effect on the relative abundance of some microbes or the opposite, if the abundance of some microbes enriched by the plant have effect on SOM values. I would say that there is some correlation, but I would not detail the cause.
- Results section: please, add information about the raw and final number of sequences per sample or per treatment and the number of OTUs. I suggest including a table with these data. E.g. see Table 1 in Kurilkina et al., (2016) (https://academic.oup.com/femsec/article/92/7/fiw094/2469997)
- Discussion section: Please, add some lines discussing the functional metagenome predictions that you show in the section 3.5.
- Suplementary tables S3-S6: please, add upper taxonomy lineage to the different taxa. E.g.: Instead of “unidentified” genus, write somehow information regarding the family, order, class or some identified taxonomic ranges. Thus, readers could have an idea of the relevance of some taxa and could compare it with their own analyses.
Reviewer 3 Report
Dear Authors,
although, as also you highlighted, the discussed topic (the effect of the intercropping practice on soil) does not have a strong originality, I think that the manuscript describe in clear way some interesting elements such as the bacterial diversity and function in the examined soils that could surely integrate the overview on the soil components affected by this practice. I believe that some minor revisions (see the attached pdf) could improve the manuscript.
except for some inconsistencies commented in the attached pdf file, I believe that the manuscript clearly discusses the experimental aspects considered (chemical and physical parameters of the soils and analysis of the bacterial community) therefore it does not require further modifications. I hope that the suggestions provided in the attached pdf can be of help in improving the manuscript.
Best regards

Author Response
Please see the attachment. Thank you for you valuable comments and suggestions.

Round 2
Reviewer 1 Report
Dear authors,
In this paper, the effects of the maize-mushroom intercropping system on soil physical-chemical properties and diversity and composition of soil microbial community are investigated in comparison to maize and mushroom monocultures.
I studied the responses. The authors give satisfactory answers to some of the comments, however, there are comments that have not been responded to or the responses are not quite satisfactory, mainly the responses that concern the Discussion section. The manuscript is still included problems as points out below.
General comments
- The changes that have been done in the Conclusion section, have not been added in the Abstract section accordingly
- In the Discussion section, there are no satisfactory explanations, embedded with the equivalent bibliography
- In the Conclusion section, the “improve soil quality” seems quite unfounded.
- The manuscript emphasizes the positive effects of intercropping of maize-mushroom, but the intercropping seems to have no important differences with mushroom monoculture. In my opinion, the authors have to reconsider the aim of this study.
Introduction
- I think that is necessary to define the “soil quality”.
Materials and Methods
- The soil texture has not been mentioned (proportion of sand, silt, clay)
- L91, I think that the word “replicates” is missing from the sentence
- The description of the experimental design has to be improved. For example: In L99 “...a small jukebox in late April …”, the April has already been referred and I suppose that the maize seeds were sowed with the same way for the maize monoculture like the intercropping system.
- The explanation for the non-parametric test is not satisfactory.
- The use of air-dried soil samples for measuring NH4-N, NO3-N is not suggested and if there is no reference that supports this methodology, the validity of data will be challenged.
Discussion
- How do the authors explain the fact that the intercropping system was not different from mushroom monoculture? The response that the authors gave: “The possible reason may be the main function of mushroom” is not sufficient.
- L302 please, remove “than” from the sentence.
- L303 “mostly due to uptake by the mushroom”, what are the mushrooms’ need for nutrients?
- L305-307 However, it seems that this happens also in mushroom monoculture (SOM of the intercropping system does not differ with SOM of mushroom).
- How do the authors explain the fact that the bacterial community diversity was higher in intercropping than maize monoculture, while the available N (meaning more food for microorganisms) was lower in the intercropping system than maize monoculture?
- L343 The word “by” is missing after stimulated
- L329-330 and L346-347, I think these sentences have the same meaning. I expected a specific conclusion in the last sentence, that deals with the above results.
- L359 “The greater abundance of Actinobacteria, Streptomyces, Nakamurella, Frateuria and Gemmatimonas indicated that intercropping promoted nutrient intake and stimulated C and N cycles” how does this conclusion arise from?
Table S8
- What does C, M and CM represent? Please add it to the legend.
Author Response
In this paper, the effects of the maize-mushroom intercropping system on soil physical-chemical properties and diversity and composition of soil microbial community are investigated in comparison to maize and mushroom monocultures. I studied the responses. The authors give satisfactory answers to some of the comments, however, there are comments that have not been responded to or the responses are not quite satisfactory, mainly the responses that concern the Discussion section. The manuscript is still included problems as points out below. Response:Thank you for your valuable comments and suggestions. We have carefully revised the texts based on all of your suggestions. Below are our point-to-point responses to your comments. General comments Point 1: The changes that have been done in the Conclusion section, have not been added in the Abstract section accordingly Response 1: The abstract section was revised accordingly. Point 2: In the Discussion section, there are no satisfactory explanations, embedded with the equivalent bibliography Response 2: We try our best to explain the results in the discussion, thank you. Point 3: In the Conclusion section, the “improve soil quality” seems quite unfounded. Response 3: The point was corrected in the new version. Point 4: The manuscript emphasizes the positive effects of intercropping of maize-mushroom, but the intercropping seems to have no important differences with mushroom monoculture. In my opinion, the authors have to reconsider the aim of this study. Response 4: Yes, the intercropping had slight or no differences with mushroom monoculture for some parameters, so our manuscript was focus on the effects of intercropping compared to maize monoculture. Introduction Point 1: I think that is necessary to define the “soil quality”. Response 1: The definition of “soil quality” was added in the Introduction. Materials and Methods Point 1: The soil texture has not been mentioned (proportion of sand, silt, clay) Response 1: The black soil is a silty clay loam classified as Mollisol. Point 2: L91, I think that the word “replicates” is missing from the sentence Response 2: The “replicates” was added to the sentence. Point 3: The description of the experimental design has to be improved. For example: In L99 “...a small jukebox in late April …”, the April has already been referred and I suppose that the maize seeds were sowed with the same way for the maize monoculture like the intercropping system. Response 3: Thanks. The maize seeds were sowed with the same way for the maize monoculture like the intercropping system. The description of the experimental design was improved. Point 4: The explanation for the non-parametric test is not satisfactory. Response 4: All data comparation was used analysis of variance (ANOVA) in IBM SPSS 23.0 software and non-parametric test was deleted from the paragraph. Point 5: The use of air-dried soil samples for measuring NH4-N, NO3-N is not suggested and if there is no reference that supports this methodology, the validity of data will be challenged. Response 5: NH4-N, NO3-N were measured by standard methods based on LY/T1228-2015. Discussion Point 1: How do the authors explain the fact that the intercropping system was not different from mushroom monoculture? The response that the authors gave: “The possible reason may be the main function of mushroom” is not sufficient. Response 1: The mushroom sticks were buried in the soil, and the samples were collected near the mushroom sticks (see Figure 1), so I think mushroom had a significantly effects. Point 2: L302 please, remove “than” from the sentence. Response 2: The “than” was removed from the sentence. Point 3: L303 “mostly due to uptake by the mushroom”, what are the mushrooms’ need for nutrients? Response 3: What I want to express is that some saprotrophs belonged to mushroom, have degradation function, which decomposed of mineral nutrients. Point 4: L305-307 However, it seems that this happens also in mushroom monoculture (SOM of the intercropping system does not differ with SOM of mushroom). Response 4: Yes, the intercropping significantly increased the contents of SOM compared with maize monoculture. Point 5: How do the authors explain the fact that the bacterial community diversity was higher in intercropping than maize monoculture, while the available N (meaning more food for microorganisms) was lower in the intercropping system than maize monoculture? Response 5: I think although the bacterial community diversity was higher in intercropping than maize monoculture, but the amount of bacteria may not be greater in intercropping than maize monoculture. Point 6: L343 The word “by” is missing after stimulated Response 6: The word “by” is added to the sentence. Point 7: L329-330 and L346-347, I think these sentences have the same meaning. I expected a specific conclusion in the last sentence, that deals with the above results. Response 7: The L329-330 emphasized the most dominant phyla in soil. The L346-347 expressed the dominant bacterial phyla could be changed by manipulating planting patterns and plant species. Point 8: L359 “The greater abundance of Actinobacteria, Streptomyces, Nakamurella, Frateuria and Gemmatimonas indicated that intercropping promoted nutrient intake and stimulated C and N cycles” how does this conclusion arise from? Response 8: The sentence was deleted in L359. Table S8 Point 1: What does C, M and CM represent? Please add it to the legend. Response 1: The C, M and CM represent maize, mushroom and intercropping, respectively. The legend was added to Table S4.